# Hypertension in Prenatally Undernourished Young-Adult Rats Is Maintained by Tonic Reciprocal Paraventricular–Coerulear Excitatory Interactions

**DOI:** 10.3390/molecules26123568

**Published:** 2021-06-11

**Authors:** Bernardita Cayupe, Carlos Morgan, Gustavo Puentes, Luis Valladares, Héctor Burgos, Amparo Castillo, Alejandro Hernández, Luis Constandil, Miguel Ríos, Patricio Sáez-Briones, Rafael Barra

**Affiliations:** 1Centro de Investigación Biomédica y Aplicada (CIBAP), Escuela de Medicina, Facultad de Ciencias Médicas, Universidad de Santiago de Chile (USACH), Santiago 9170020, Chile; bernardita.cayupe@usach.cl; 2Laboratorio de Neurofarmacología y Comportamiento, Escuela de Medicina, Facultad de Ciencias Médicas, Universidad de Santiago de Chile, Santiago 9170020, Chile; carlos.morgan@gmail.com (C.M.); gustavoadolfopuentesgarrido@gmail.com (G.P.); 3Laboratorio de Hormonas y Receptores, Instituto de Nutrición and Tecnología de los Alimentos (INTA), Universidad de Chile, Santiago 7830490, Chile; lvallada@inta.uchile.cl; 4Escuela de Psicología, Facultad de Ciencias, Universidad Mayor, Santiago 7570008, Chile; hector.burgosg@mayor.cl; 5Escuela de Psicología, Facultad de Ciencias Sociales y Comunicaciones, Universidad Santo Tomás, Santiago 8370003, Chile; amparocastillobo@santotomas.cl; 6Laboratorio de Neurobiología, Departamento de Biología, Facultad de Química y Biología, Universidad de Santiago de Chile, Santiago 9170020, Chile; alejandro.hernandez@usach.cl (A.H.); luis.constandil@usach.cl (L.C.); miguel.rios@usach.cl (M.R.)

**Keywords:** hypertension, prenatal undernutrition, paraventricular nucleus, locus coeruleus, noradrenaline, corticotropin-releasing factor, α_1_-adrenoceptor, rat

## Abstract

Prenatally malnourished rats develop hypertension in adulthood, in part through increased α_1_-adrenoceptor-mediated outflow from the paraventricular nucleus (PVN) to the sympathetic system. We studied whether both α_1_-adrenoceptor-mediated noradrenergic excitatory pathways from the locus coeruleus (LC) to the PVN and their reciprocal excitatory CRFergic connections contribute to prenatal undernutrition-induced hypertension. For that purpose, we microinjected either α_1_-adrenoceptor or CRH receptor agonists and/or antagonists in the PVN or the LC, respectively. We also determined the α_1_-adrenoceptor density in whole hypothalamus and the expression levels of α_1A_-adrenoceptor mRNA in the PVN. The results showed that: (i) agonists microinjection increased systolic blood pressure and heart rate in normotensive eutrophic rats, but not in prenatally malnourished subjects; (ii) antagonists microinjection reduced hypertension and tachycardia in undernourished rats, but not in eutrophic controls; (iii) in undernourished animals, antagonist administration to one nuclei allowed the agonists recover full efficacy in the complementary nucleus, inducing hypertension and tachycardia; (iv) early undernutrition did not modify the number of α_1_-adrenoceptor binding sites in hypothalamus, but reduced the number of cells expressing α_1A_-adrenoceptor mRNA in the PVN. These results support the hypothesis that systolic pressure and heart rate are increased by tonic reciprocal paraventricular–coerulear excitatory interactions in prenatally undernourished young-adult rats.

## 1. Introduction

Prenatal undernutrition can program various adaptations in humans and experimental animals. These adaptations, in the short-term, may be beneficial for fetal survival under stress conditions. However, in the long-term, they can lead to metabolic imbalances and cardiovascular diseases, such as hypertension. It has been proposed that maladaptive neuroendocrine programming install chronic hypertension during adulthood in prenatally undernourished subjects [1,2,3]. Animals subjected to prenatal malnutrition showed elevated expression levels of hypothalamic corticotropin-releasing factor (CRF) after birth, as well as chronically increased plasma levels of both adrenocorticotropin (ACTH) [4] and corticosterone hormones [5,6]. These data suggest increased activity of the hypothalamus–pituitary–adrenal (HPA) axis in malnourished animals. Besides, studies in adult men and women showed that low birthweight—an index of fetal malnutrition—was associated with activation of the HPA axis, high fasting cortisol levels, and increased cortisol response evoked by exogenous ACTH [6]. Together with these functional and chemical changes observed in early malnourished subjects, an increase in spontaneous neuronal activity can be observed in the paraventricular nucleus (PVN) of rats subjected to prenatal malnutrition [7]. The PVN is important in regulating the sympathetic output and blood pressure [8,9]. Thus, both the hypophyseal–adrenal axis and the sympatho-adrenomedullary system constitute output mechanisms whereby neurons of the PVN would trigger hypertension in previously undernourished subjects. Through these mechanisms, the PVN neurons can control peripheral cardiovascular viscera via corticosterone secretion and noradrenaline release.

Rats undergoing prenatal malnutrition show reduced glucocorticoid receptor (GR) expression in the hypothalamus throughout later juvenile and adult life [10], minimizing the feedback control of the HPA axis, keeping it hyperactive in adulthood. This reduced GR expression has also been observed in the pituitary gland of intrauterine undernourished ovine fetuses [11]. Since the activity of placental 11β-hydroxysteroid dehydrogenase type 2—the enzyme that catalyzes rapid conversion of cortisol and corticosterone to inert steroids—is significantly decreased in undernourished rat fetuses [10,12], it seems likely that reductions of GR expression in the HPA axis can result from fetal overexposure to maternal glucocorticoids. However, other studies failed to find decreased GR mRNA in the PVN [11] or in the whole hypothalamus [13] of prenatally undernourished ewes and rats [14]. On the contrary, Stevens et al. [15] observed increased GR mRNA levels in the hypothalamus of perinatally undernourished ewes. Additionally, Li et al. [16] found that GR immunoreactivity was unchanged in the PVN of pre-term baboons removed by cesarean surgery at 90% gestation from mothers fed 70% of the diet eaten by ad libitum fed controls females, despite that CRF peptide expression in the PVN and peripheral ACTH and cortisol were significantly increased. Those data suggest that the elevated activity of the HPA axis exhibited by prenatally malnourished subjects is initiated by an increase of CRF levels in the hypothalamus, which appears to develop independently of changes in GR expression in the PVN.

Most likely, increased CRF expression induced by prenatal malnutrition results from central NA hyperactivity developed in these animals shortly after birth. Brain of prenatally malnourished rats exhibits increased synthesis, release, and noradrenaline turnover [17,18,19,20,21,22,23,24,25,26], a neurotransmitter that stimulates CRF transcription in the PVN [27] by activating α_1_-adrenoceptors [28], and a subsequent increase in cytosolic calcium levels in PVN neurons [29] and CRF secretion [30]. Dense axonal projections from the A1, A2, A6 (*Locus coeruleus*, LC) and A7 noradrenergic groups of the brainstem innervate the magnocellular and parvocellular regions of the PVN [31,32,33]. Additionally, electrical stimulation of this brainstem–PVN connection excites the majority of PVN neurons, an effect that is counteracted by the α_1_-adrenoceptor antagonist, ergotamine and mimicked by the α_1_-adrenoceptor agonist, phenylephrine [34,35,36]. Importantly, prenatally malnourished rats showed greater (near twice) basal neuronal activity both in the LC and the PVN compared to normal eutrophic animals, which was reduced in the two nuclei to baseline levels after intra-PVN microinjection of the α_1_-adrenoceptor antagonist prazosin [7]. This was interpreted as these two nuclei being reciprocally interconnected by excitatory neurons, which influence each other in animals with prenatal malnutrition [7]. In fact, besides the mentioned excitatory α_1_-adrenoceptor-mediated brainstem to PVN connections, there are reciprocal CRFergic axonal projections from PVN to LC [37,38], which excite LC neurons [39,40] through CRF1 receptors [41,42]. Thus, an excitatory LC-PVN closed-loop can keep PVN neurons activated in prenatally malnourished animals and then contribute to a sustained activity of the sympathetic system, generating hypertension. This hypothesis is supported by the following facts: (i) CRF receptors have been unequivocally detected in noradrenergic LC neurons, as revealed by CRF labeling in perikarya and dendrites of tyrosine hydroxylase-positive LC neurons in dual-labeling observations under both epifluorescence and electron microscopy [43,44]; (ii) a combined immunohistochemical-autoradiographic approach at the electron microscopic level has demonstrated the existence of α_1_-adrenoceptors in CRF-immuno-positive perikarya in the parvocellular PVN of adult rats [45]; (iii) cell-attached recording in retrogradely labeled and electrophysiologically identified-PVN neurons able to express the CRF mRNA showed that noradrenergic excitation could be blocked by the α_1_-adrenoceptor antagonist prazosin and mimicked by the α_1_-adrenoceptor agonist phenylephrine [46]. These data strongly suggest that α_1_-adrenoceptors are located in CRF-expressing-parvocellular neurons in the PVN.

In a similar manner, the feed-forward activation between the LC and the PVN lead to some disorders related to stress and anxiety [47,48]. It should be noted, however, that only the LC-PVN arm [7], but not the reciprocal PVN-LC arm of the proposed bidirectional neuronal communication between these two complementary nuclei had been tested until now in prenatally undernourished adult rats. The current study addresses the bidirectional contribution of reciprocal excitatory interactions from LC to PVN and from PVN to LC in the production of hypertension in prenatally malnourished adult rats, along with testing whether any type of sensitization occurs in the PVN, the exit point of the circuit.

To functionally characterize the dependence of cardiovascular parameters on the integrity of the proposed tonic PVN–LC bidirectional loop, we tested the effects of microinjecting either a specific α_1_-adrenoceptor antagonist (prazosin) or agonist (phenylephrine) into the PVN, and/or either an antagonist (a-helical CRF) or an agonist (CRF) of CRF receptor into the LC, on the systolic blood pressure and the heart rate of both eutrophic and prenatally undernourished adult offspring. Antagonists administered in any of those complementary nuclei should disrupt the loop and normalize both blood pressure and heart rate, while agonists should exert the opposite effect by reinforcing the loop. Furthermore, the facilitating cardiovascular effect of an agonist in any of both nuclei should be counteracted by microinjecting the corresponding antagonist into the complementary nucleus in prenatally undernourished adult rats. Moreover, since maintenance of a feed-forward loop would require some type of sensitization at the output point of the circuit, in this case the PVN, we measured the changes in hypothalamic α_1_ -adrenoceptor density as well as the expression levels of α_1_-adrenoceptor mRNA in the PVN in both prenatally malnourished animals and eutrophic controls.

## 2. Results

### 2.1. Effect of Maternal Dietary Treatment on the Body and Brain Weights, and the Cardiovascular Parameters in the Young-Adult Progeny

The body (** *p* < 0.01) and brain (*** *p* < 0.001) weights of malnourished rats (Table 1) were significantly lesser than eutrophic rats, whereas significant increases in systolic arterial pressure (** *p* < 0.05) and heart rate (* *p* < 0.05) were observed in the malnourished animals. Data were analyzed by a two-tailed unpaired Student’s *t*-test, and no statistically significant differences were observed between groups in diastolic pressure. Every α_1_-adrenergic ligand tested produced quite similar results in systolic pressure and heart rate (see below), whereas the microinjection of aCSF in either of the two nuclei did not show any effect (Figure 1, *p* > 0.05, two-way repeated measures ANOVA followed by intragroup and intergroup statistics using Dunnett or Bonferroni multiple comparisons test, respectively, *n* = six rats in each group). Along with serving as a control for eventual unwanted effects associated with the microinjection procedure itself, this result provides an exact representation of the measured hypertensive process (over a 50 min period) developed in prenatally undernourished adult subjects.

### 2.2. Cardiovascular Effects of Bilateral Intra-PVN Microinjection of α_1_-Adrenergic Ligands

Figure 2A shows that the systolic arterial pressure in the malnourished group was significantly higher than that of the normal group (^##^*p* < 0.01, intergroup comparisons, two-way repeated measures ANOVA followed by Bonferroni’s multiple comparisons test, *n* = six rats in each group) even at t = 0 min, i.e., before any drug administration procedure. Bilateral microinjection of 1 µg of the α_1_-adrenoceptor agonist phenylephrine into both PVNs significantly increased systolic pressure in normal rats (* *p* < 0.05, at 20 and 30 min after microinjection, intragroup comparisons, two-way repeated measures ANOVA followed by Bonferroni multiple comparisons test). In contrast, systolic pressure was not modified by phenylephrine in the undernourished group. Intergroup comparisons revealed that during the 10 to 40 min interval after phenylephrine, the systolic pressure scores from normal rats were not statistically different from their undernourished hypertensive counterparts (*p* > 0.05, two-way repeated measures ANOVA followed by Bonferroni multiple comparisons test). Figure 2B shows a significant decrease of the systolic pressure in the undernourished group after bilateral intra-PVN microinjection of 50 ng prazosin (** *p* < 0.01 at 20 min, and * *p* < 0.05 at 30 min, *n* = six rats in each group), thereby reaching similar pressure scores to those observed in the normal group. In contrast, prazosin microinjection into both PVNs did not modify systolic pressure in normal rats (*p* > 0.05, two-way repeated measures ANOVA followed by Bonferroni multiple comparisons test, n = six rats in each group). Figure 2C shows that before administering noradrenergic ligands, undernourished rats had significantly increased heart rate (^##^
*p* < 0.01, intergroup comparisons, two-way repeated measures ANOVA followed by Bonferroni multiple comparisons test, *n* = six rats in each group) compared to normal animals. Following bilateral intra-PVN phenylephrine microinjection, heart rate increased in normal animals (* *p* < 0.05, at 20 and 30 min, intragroup comparisons, two-way repeated measures ANOVA followed by Bonferroni’s multiple comparisons test) but not in the undernourished group. In contrast, prazosin microinjection into both PVNs (Figure 2D) significantly decreased systolic pressure in malnourished subjects (at least * *p* < 0.05, 10, 20, 30 and 40 min after microinjection, *n* = six rats in each group), while it did not produce any effect on the heart rate of eutrophic animals. Of note, the α_1_-adrenergic agonist used (phenylephrine) was effective in increasing both systolic pressure and heart rate only in normal animals while producing no effect in the undernourished counterpart; by contrast, the α_1_-adrenergic antagonist (prazosin) was effective in lowering systolic pressure and heart rate only in malnourished animals, while it was devoid of effects in normal animals.

### 2.3. Cardiovascular Effects of Bilateral Intra-LC Microinjection of CRFergic Ligands

Figure 3A shows that 20 pmol CRF microinjected into both LCs rapidly increased the systolic pressure recorded from normal rats (*** *p* < 0.001 at 10 min, and ** *p* < 0.01 at 20 min after microinjection, intragroup comparisons, two-way repeated measures ANOVA followed by Dunnett’s multiple comparisons test, *n* = six rats), without producing any effect in the undernourished group. Comparisons between groups (Bonferroni’s multiple comparisons test, *n* = six rats in each group) show that when the effect of CRF peaked (at 10 and 20 min after CRF microinjection), systolic pressure scores in normal rats were not statistically different from those in malnourished hypertensive animals. Figure 3B shows that bilateral intra-LC microinjection of 26 pmol of the CRF receptor antagonist α-helical CRF significantly decreased systolic pressure in the malnourished group (** *p* < 0.01 at 10 and 20 min after microinjection, *n* = six rats in each group). In contrast, α-helical CRF did not modify the systolic pressure in normal animals. Figure 3C shows that bilateral intra-PVN microinjection of CRF in normal animals increased the heart rate (** *p* < 0.01 at 10 min after microinjection, intragroup comparisons using two-way repeated measures ANOVA followed by Dunnett’s multiple comparisons test, *n* = six rats), but it had no effect in undernourished rats. Conversely, α-helical CRF microinjected into both LCs of malnourished rats (Figure 3D) decreased the heart rate (** *p* < 0.01 at 10 min after microinjection, *n* = six rats), while it did not produce any effect on the heart rate of normal animals. Again, the agonist (CRF) effectively increased systolic pressure and heart rate only in normal animals without producing any effect in malnourished ones, while the antagonist (α-helical CRF) lowered systolic pressure and heart rate only in the malnourished group, showing no effects on eutrophic rats.

### 2.4. Cardiovascular Effects of Intra-PVN or Intra-LC Agonist Microinjection, as Modified by Microinjecting the Corresponding Antagonist into the Complementary Nucleus

Figure 4 shows that the enhancing effect of microinjecting 100 ng CRF into both LCs upon systolic pressure (Figure 4A, ** *p* < 0.01 at 5 min, *** *p* < 0.001 at 10 and 15 min, and * *p* < 0.05 at 20 min after microinjection) and heart rate (Figure 4B, *** *p* < 0.001 at 5 min, ** *p* < 0.01 at 10 min, and * *p* < 0.05 at 15 min after microinjection) in normal eutrophic rats was almost unchanged by a bilateral intra-PVN 50 ng prazosin microinjection performed 10 min before CRF administration (intragroup comparisons using two-way repeated measures ANOVA followed by Dunnett’s multiple comparisons test, *n* = six rats). Apart from a shorter (but not lower) effect, the time-course for the impact of intra-CL CRF on the cardiovascular measures is very similar to that observed in control eutrophic rats with intact LC-PVN noradrenergic connection (see Figure 2A,C), indicating that in eutrophic animals, the α_1_-adrenergic connection from the LC to the PVN is not required for CRF to exert its incremental effect on the measured cardiovascular parameters. The results also showed that the increase in systolic pressure (Figure 4C, * *p* < 0.05 at 5 and 15 min, and ** *p* < 0.01 at 10 min after microinjection) and heart rate (Figure 4D, ** *p* < 0.01 at 5 min, and * *p* < 0.05 at 10 min after microinjection) induced by intra-PVN bilateral infusion of 1 µg of the α_1_-adrenoceptor agonist phenylephrine in eutrophic controls was not prevented by microinjecting 100 ng α-helical CRF into the LC (intragroup comparisons using two-way repeated measures ANOVA followed by Dunnett’s multiple comparisons test, *n* = six rats), which indicates that the CRFergic connection of the PVN to the LC is not involved in the increases of systolic pressure and heart rate that phenylephrine induces when administered into the PVN of healthy rats.

It should be noted that although the undernourished hypertensive rats did not respond to the administration of agonists either in the PVN (see Figure 2A,C) or in the CL (see Figure 3A,C), bilateral microinjection of prazosin into the PVN of these animals together with producing a reduction in blood pressure and heart rate to normal levels (Figure 4A,B, ^####^
*p* < 0.0001 and ^###^
*p* < 0.001, respectively, intergroup comparisons using two-way repeated measures ANOVA followed by Bonferroni’s multiple comparisons test, *n* = six rats in each group), allowed for malnourished rats could fully regain their responsiveness to the intra-LC CRF challenge. Indeed, microinjection of 100 ng CRF into the LC of prazosin-pretreated undernourished animals induced increases of both systolic pressure (Figure 4A, * *p* < 0.05 at 10 and 20 min, ** *p* < 0.01 at 15 min, and *** *p* < 0.001 at 25 and 30 min after microinjection) and heart rate (Figure 4B, * *p* < 0.05 at 5 min, ** *p* < 0.01 at 10 and 25 min, and *** *p* < 0.001 at 15, 20 and 30 min after microinjection) to levels showed by the animals before prazosin administration (assessed by two-way repeated measures ANOVA followed by Dunnett’s multiple comparisons test, *n* = six rats). Similarly, 100 ng α-helical CRF microinfused into both LCs rescued undernourished rats from hypertension and tachycardia (Figure 4C,D, ^#^
*p* < 0.05, intergroup comparisons using two-way repeated measures ANOVA followed by Bonferroni’s multiple comparisons test, *n* = six rats in each group) and allowed the rats to recover responsiveness to phenylephrine administration into the complementary PVN nucleus. Indeed, phenylephrine microinfused into the PVN of α-helical CRF-pretreated undernourished animals induced increases of both systolic pressure (Figure 4C, * *p* < 0.05 at 15 and 30 min, ** *p* < 0.01 at 20 and 25 min after microinjection) and heart rate (Figure 4D, * *p* < 0.05 at 5 min, ** *p* < 0.01 at 10 min, *** *p* < 0.001 at 15, 20, 25 and 30 min after microinjection). Therefore, intra-PVN α_1_-adrenergic agonists and intra-LC CRFergic agonists are ineffective in modifying cardiovascular parameters in malnourished rats insofar as they are hypertensive. However, when blood pressure and heart rate in these animals are reduced near the normal level by microinjection of an appropriate antagonist into the complementary nucleus, the agonists regain their full potential to induce hypertension and tachycardia. This experimental series indicates that the ability of CRF and phenylephrine to increase systolic pressure and heart rate when administered into the appropriate nucleus is highly dependent on the animal’s cardiovascular condition so that those drugs are inactive in malnourished hypertensive rats but highly active in normotensive controls and in malnourished rats rescued from their state of hypertension.

### 2.5. Total α_1_-Adrenoceptor Binding in the Hypothalamus

In normal 40 day-old rats, the concentration-dependent binding of [^3^H]-prazosin to the hypothalamus membranes resulted in a linear Scatchard plot. Analysis of the binding data revealed a Kd = 0.127 ± 0.028 nM and a Bmax = 136.3 ± 13.9 fmol/mg protein for the total α_1_-adrenoceptor population in the whole hypothalamus. Comparatively, in the hypothalamus of prenatally malnourished rats of similar age, [^3^H]-prazosin binding showed a statistically similar Kd = 0.121 ± 0.027 nM and Bmax = 130.7 ± 12.8 fmol/mg protein (*p* > 0.05, unpaired two-tailed Student’s *t*-test), thus indicating that the total number of α_1_-adrenoceptor binding sites (α_1A_ + α_1B_ + α_1D_) was unaltered in the hypothalamus of malnourished rats (Figure 5).

### 2.6. Expression of α_1A_-Adrenoceptor mRNA in the PVN

The α_1A_ probe hybridized moderately with neurons in the PVN of normal eutrophic rats (Figure 6A, left panel), while the unlabeled control probe originated only background staining (not shown). Prenatally malnourished 40-day old rats had a lower number of stained cells expressing α_1A_-adrenoceptor mRNA in the PVN than normal eutrophic rats of the same age, as can be observed in Figure 6A (right panel vs. left panel). Quantification by pixel counting with image J software allowed the calculation of a 66% reduction of cells expressing α_1A_-adrenoceptor mRNA in the PVN of malnourished rats (106 ± 14 vs. 166 ± 16, *p* < 0.05, two-tailed unpaired Student’s *t*-test) as compared to eutrophic controls (Figure 6B).

## 3. Discussion

Measurements performed in young-adult rats of 40 days of age revealed substantial body and brain weight deficits in those animals that had been malnourished during fetal life. As reported elsewhere, prenatal malnutrition results in long-lasting brain weight deficits through mechanisms involving loss of neurons, glial cells and myelin content, impaired dendritic differentiation, and maladaptive epigenetic programming [50,51]. At days 40 to 44 of age, significant increases both in systolic pressure and heart rate were observed in prenatally malnourished rats, which agree with previous reports showing that the progeny of normotensive rats subjected to maternal undernutrition during pregnancy develop hypertension [12,52,53] and exhibit increased heart rate [53,54] at adulthood. Similar increases in arterial blood pressure and heart rate have been observed in the adult progeny of perinatally (gestation plus lactation periods) undernourished rats [55], associated with an increased cardiovascular sympathetic tone [56].

Notably, the foregoing results show that microinjecting agonists into the appropriate nucleus (phenylephrine into the PVN, CRF into the LC) increased both systolic pressure and heart rate in normotensive eutrophic rats but they were ineffective when administered to the malnourished group. Microinjecting appropriate antagonists in the respective nuclei (prazosin into the PVN and α-helical CRF into the LC) reduced both the elevated systolic pressure and heart rate shown by undernourished rats but the antagonists failed to modify any cardiovascular parameters in eutrophic controls. Additionally, the increased systolic pressure and heart rate induced by agonist microinjection in normotensive eutrophic rats could not be prevented by blocking the synaptic communication to the complementary nucleus with the appropriate antagonist. However, in undernourished rats, administration of an antagonist in one of the nuclei effectively allowed the complementary nucleus to fully respond to the agonist administered, inducing hypertension and tachycardia. Finally, early undernutrition did not modify the number of α_1_-adrenoceptor binding sites in hypothalamus but reduced the number of cells expressing α_1A_-adrenoceptor mRNA in the PVN.

These results are analyzed taking into account (i) the roles that PVN and LC play in cardiovascular regulation and the receptors involved, (ii) the effectiveness of antagonists only in malnourished rats and of agonists only in eutrophic rats, and (iii) the hypothesis of a tonically active reciprocal neural communication between PVN and LC in prenatally malnourished adult subjects, but not in eutrophic controls.

### 3.1. Simultaneous Tonic Release of Noradrenaline in the PVN and CRF in the LC Is Involved in Hypertension and Increased Heart Rate in Prenatally Malnourished Rats, but Not in the Baseline Cardiovascular Values of Eutrophic Controls

The present results showed that systolic pressure and heart rate, which were found to be increased in the malnourished group, fell rapidly (but transiently) near control values after either intra-PVN administration of the α_1_ adrenergic receptor antagonist (prazosin) or intra-LC injection of the antagonist for CRF receptors (α-helical CRF). These observations agree with a previous report showing decreases in arterial pressure and heart rate after intra-PVN prazosin micro-infusion in undernourished hypertensive rats [48] and also with the antihypertensive effect of i.c.v. α-helical CRF [57] or antalarmin, a CRF receptor antagonist [58] in different rat models of hypertension. Prazosin is a selective inverse agonist for α_1_-adrenoceptors [59] that binds almost equally the α_1A_, α_1B_, and α_1D_ adrenoceptor subtypes [60], while α-helical CRF is a non-selective competitive antagonist for the three CRF receptor subtypes known in the rat, CRF1, and the splice variants CRF_2(a)_ and CRF_2(b)_ [61]. As competitive ligands, their ability to counteract receptor-mediated effects (e.g., elevated cardiovascular scores in malnourished animals) requires from excitation of the receptors by tonically released endogenous agonists, namely, noradrenaline in the PVN and CRF in the LC.

Remarkably, either intra-PVN-prazosin or intra-LC-α-helical CRF, independently, did rescue prenatally undernourished rats from hypertension and tachycardia. This evidence suggests that simultaneous concurrent tonic neuronal activity from both nuclei is required to maintain elevated scores of systolic pressure and heart rate in prenatally undernourished animals. Mechanistically, the occurrence of simultaneous tonic neuronal hyperactivity in both the PVN and the LC of prenatally undernourished adult rats could reflect independent events in each nucleus (both caused by prenatal undernutrition), or synaptically mediated interdependent events, where the increased neural activity in one of the nuclei (caused by undernutrition) is transferred to the other nucleus by neural connections. The first alternative is partly supported by data showing increased potassium-evoked norepinephrine release in the cerebral cortex of undernourished rats [20,21,24,26], which is released only from axons originated in the LC [62], but similar determinations are lacking in the hypothalamus of undernourished animals. Indeed, noradrenaline turnover is enhanced in the hypothalamus of prenatally undernourished rats [23], but the release of noradrenaline in the hypothalamus was not determined in that study. Besides, although undernutrition during fetal life has been found to result in increased neuronal activity [7] and higher CRF expression [4] in the PVN during adulthood, it has not yet been determined whether these events could translate themselves into enhanced release of CRF in the CL through the PVN-CL CRFergic pathway. The second alternative shares the same supporting evidence and drawbacks as the first, and it is based on the existence of reciprocal excitatory pathways between both nuclei: PVN to LC CRFergic connections [39,40,41,42], and LC to PVN α_1_-adrenergic connections [34,35,36,63]. This alternative is consistent with studies carried out in other chronic syndromes, such as chronic stress, which show that in such a situation there is greater noradrenergic stimulation of the PVN [64] along with increased CRFergic stimulation of the LC [65]. As noted by Dunn et al. [48], in some conditions (stress, panic), excitatory reciprocal interaction between LC and PVN neurons could potentially lead to a vicious cycle with the mutual activation escalating the activity of noradrenergic and CRFergic systems in an uncontrolled manner, which also might be occurring in neurons of both LC and PVN of prenatally undernourished rats, as suggested previously [7]. Therefore, the present results support that simultaneous concurrent tonic neuronal activity in both nuclei—the PVN and LC—is required to maintain elevated scores of systolic pressure and heart rate in the prenatally malnourished animals, as any of the two antagonists (prazosin into the PVN, α-helical CRF into the LC) could independently alleviate both hypertension and tachycardia in rats with early undernutrition.

The current study has also shown that the α_1_-adrenoceptor antagonist prazosin and the CRF receptor antagonist α-helical CRF did not produce any cardiovascular effect when microinfused into the PVN or the LC, respectively, of eutrophic controls, in sharp contrast with the marked hypotensive and anti-tachycardia effects induced by the same drugs when administered to previously malnourished animals. Coherently, previous data indicated that intra-PVN prazosin microinjection did not modify neither the arterial pressure nor the heart rate in normotensive animals [66]. Conversely, the present results showed that administration of the respective agonists (phenylephrine intra-PVN or CRF intra-LC) increased systolic pressure and heart rate in eutrophic normotensive animals, though they were ineffective in modifying these cardiovascular parameters in the prenatally malnourished group.

The question then arises why the mentioned antagonists depressed arterial pressure and heart rate in malnourished–hypertensive rats but not in eutrophic ones, whereas the agonists were only effective in eutrophic–normotensive animals. Two non-excluding possibilities may help to account for these observations: (i) in prenatally undernourished animals, both the α_1_-adrenergic and CRF1 receptors could undergo desensitization in neurons of the PVN and the LC, respectively, due to the hyperactivity exhibiting the central noradrenergic [17,20,22,23,24,25,26] and CRFergic [4,16] systems in those animals; and/or (ii) in prenatally undernourished animals the PVN and LC neurons are already fully active and therefore insensitive to further excitation by application of exogenous agonists. Although both α_1_- adrenoceptors [67] and CRF receptors [68] could undergo desensitization in the brain (e.g., after exposure to chronic stress), the first alternative seems unlikely because receptor desensitization should decrease the effectiveness of the respective agonist but also that of the antagonist, and in the present study, the malnourished rats were insensitive to agonists but not to antagonists. The second alternative is supported by data showing that the LC and PVN basal neuronal activity in drug-free undernourished rats is near twice that of normal rats [7,69], which possibly make it unlikely that both types of neurons can be further excited by exogenous drugs. Indeed, in naïve healthy rats, the spontaneous discharge rate of LC neurons could not be enhanced by more than two-fold after high doses of intra-LC [39] or intracerebroventricular [39,70] CRF, an activity level roughly mimicking that of the malnourished rat. The fact that both antagonists failed to reduce cardiovascular scores in eutrophic controls leads to the notion that in the PVN and LC of normal rats, there is no such tonic release of norepinephrine or CRF or, at least, that in healthy rats, the tonic release of norepinephrine and CRF are not significantly involved in the baseline values of systolic pressure and heart rate. Hence, the present results are consistent with the idea that a specific subset of neurons in the PVN and the LC, involved in the increases of arterial pressure and heart rate, are fully activated in prenatally malnourished rats by the tonic release of the respective local excitatory neurotransmitters, namely noradrenaline and CRF. At the same time, they did not support such a tonic release of noradrenaline and CRF in the PVN and LC of eutrophic controls.

### 3.2. Tonic Release of Norepinephrine in the PVN and Tonic Release of CRF Are Synchronized by Reciprocal or Serial Excitatory Connections between the PVN and the LC in Prenatally Malnourished Rats

Importantly, once the elevated systolic pressure in the malnourished rats was lowered close to the normal level with one of the antagonists used herein (e.g., intra-PVN microinjected prazosin), the agonist administered in the complementary nucleus (e.g., intra-LC microinjected CRF) recovered its full potential to induce hypertension and tachycardia. This can be interpreted as that the inhibition of PVN neurons by the α_1_-adrenoceptor antagonist prazosin, which relieves PVN neurons from the tonic excitatory influence of noradrenergic hyperactivity existing in the brain of early malnourished animals [20,21,24,25,26], results in a secondary downstream relief of LC neurons from the tonic excitatory influence of endogenous CRF released by axonal terminals arising from the PVN, leaving these neurons fully responsive to exogenously administered CRF. This result is consistent with already reported data showing that the microinjection of prazosin into the PVN of previously undernourished rats induced significant decreases of neuronal activity in the PVN but also in the LC [7], and strongly argues in favor of the existence of functional excitatory CRFergic connections from the PVN to the LC in prenatally malnourished rats. A similar picture emerged when hypertension in malnourished rats was challenged with intra-LC microinjected α-helical CRF. In fact, phenylephrine administered intra-PVN did not induce any change in systolic pressure and heart rate in previously malnourished adult hypertensive rats, but prior microinjection of α-helical CRF into the LC of these animals allowed intra-PVN phenylephrine to recover its effectiveness to induce hypertension and tachycardia. This observation implies that relieving of LC noradrenergic neurons from the tonic excitatory influence of CRF in early malnourished animals by the CRF receptor antagonist α-helical CRF resulted in downstream alleviation of PVN neurons from the tonic excitatory influence of endogenous noradrenaline released by axonal terminals coming from the LC, leaving PVN neurons fully responsive to exogenously administered phenylephrine. Electrophysiological studies currently in progress in our laboratory support this contention by showing that microinjection of α-helical CRF into the LC of prenatally undernourished adult rats shut down the increased spontaneous neuronal activity both in the LC and the PVN, almost simultaneously (unpublished observations). Therefore, it is likely that both the CRFergic connections from the PVN to the LC and the noradrenergic pathway from the LC to the PVN play an important role in the synchronization of the increased neuronal rhythm between the PVN and the LC in malnourished animals, as well as in the determination of the downstream activity of the sympathetic system with the appearance of hypertension and tachycardia in these animals. Taken together, these results support the notion that reciprocal, tonically active excitatory connections between the PVN and the LC are involved in the generation of hypertension and increased heart rate in previously malnourished adult rats. In contrast, these reciprocal connections between the PVN and the LC seem to be devoid of tonic activity in healthy normotensive rats, since blood pressure and heart rate in these animals were insensitive to microinjection of the appropriate antagonist in the PVN or the LC. Those results are depicted schematically in Figure 7A.

Such a tonically active, reciprocal excitatory feed-forward loop between the PVN and the LC that is observed in prenatally undernourished adult subjects is consistent with the organization and function of local neural circuits in both the PVN and the LC, which include some modulatory interneurons that have been well-characterized [46,71,72,73,74]. Indeed, in addition to the direct excitatory α_1_-adrenoceptor-mediated noradrenergic input in CRF-expressing parvocellular neurons in the PVN (29,30,34–36), it is known that the activity of these neurons is synaptically modulated by GABA-synthesizing interneurons surrounding the PVN [71]. A decrease in the inhibition exerted by these interneurons has been reported to result in increased sympathetic activity triggered from the PVN [72]. In close agreement, noradrenergic input has been shown to inhibit PVN GABAergic interneurons via α_2_ adrenoceptors, resulting in disinhibition of parvocellular PVN neurons [46,73]. α_2_-adrenoceptor-mediated disinhibition of PVN neurons from GABAergic interneurons control would clearly reinforce the α_1_-adrenoceptor-mediated direct activating effect on parvocellular neurons of the PVN. In addition, α_1_-adrenergic stimulation of intra PVN local glutamatergic interneurons can also activate the parvocellular PVN neuronal population in a parallel heterosynaptic fashion [75], giving rise to hypertension and tachycardia. A fairly similar picture emerges when considering modulatory influences on LC by axons from other brain regions and from local interneurons. Indeed, in addition to the excitatory CRF input received by LC neurons that synthesize norepinephrine [38,43,44], these neurons show synaptic contacts from a GABAergic population of interneurons found in the peri-LC region, which provide local regulation mediated by the GABA_A_ receptor upon the activity of noradrenergic neurons of the LC [74,76]. Studies using a combination of anatomical, electrophysiological and optogenetic tools have revealed that noradrenergic LC-core neurons and GABAergic peri-LC neurons receive axon afferents from different brain regions concerned with processing and evaluation of sensory stimuli and stressors, which are involved in the regulation of waking/arousal and a diversity of state-dependent cognitive and motivational processes. Nevertheless, the input coming from the PVN arrived rather directly to noradrenaline-expressing LC-core neurons [76], thus supporting a direct PVN-LC connection as part of the feed-forward excitatory loop that appears in prenatally undernourished adult rats (Figure 7B).

### 3.3. PVN as the Output System That Mediates Hypertension and Increased Heart Rate during Tonic Activation of PVN and LC Neurons in Prenatally Malnourished Rats

We have previously defined the PVN as the output system mediating hypertension and increased heart rate during tonic activation of PVN and LC neurons in prenatally malnourished rats, because the PVN provides a dominant source of excitatory drive to the cardiovascular system via sympathetic outflow (for reviews see references [9,77,78]). Indeed, parvocellular CRFergic neurons [79] alongside other peptidergic parvocellular neurons [9,78] project densely to the rostral ventrolateral medulla, which are known to establish direct synaptic relationships with spinal preganglionic sympathetic neurons that control the sympathetic output to different target organs involved in the regulation of blood pressure, including the heart and blood vessels, the kidneys, and the adrenal medulla (Figure 7B). In contrast, the LC is a noradrenergic integrative center that distributes stress-related afferent signals throughout the forebrain structures (cerebral cortex, hippocampus, cerebellum, most thalamic nuclei, and partially to the hypothalamus). Although there are data showing that the LC can modulate blood pressure and heart rate, LC activation is known to give rise to mixed vasopressor and vasodepressor effects [62].

If in previously malnourished rats the PVN is the dominant output system that downstream transfers the neuronal hyperactivity of the PVN and LC into hypertension and tachycardia, such a system should at least exhibit the two following properties: (i) it must be devoid of desensitization to be able to operate full time; in particular, a negative regulation of α_1_-adrenergic receptors should not be observed in the PVN of these animals; and (ii) maintenance of neuronal activity in such a feed-forward loop would require some type of long-lasting sensitization at the loop exit point, namely in the PVN. The first statement is consistent with the neurochemical data obtained in the present study, since early undernutrition did not modify the number of α_1_-adrenoceptor binding sites in the hypothalamus as related to eutrophic controls. However, the PVN of these animals exhibited a reduced number of cells expressing α_1A_-adrenoceptor mRNA. Different factors may account for this apparent discrepancy: (i) the binding assay performed involved tissue from the entire hypothalamus, a technical limitation originating in the amount of tissue required for the determinations, while in situ hybridization allowed detection of mRNA in specific restricted hypothalamic region, such as the PVN; (ii) [3H]-prazosin binding identified the three α_1_-adrenoceptor subtypes, because prazosin binds, almost equally, the α_1A_, α_1B_, and α_1D_ adrenoceptor subtypes [60], whereas the deoxynucleotide probe for in situ hybridization was specific for the α_1A_-adrenoceptor mRNA subtype; (iii) although the three α_1_-adrenoceptor subtypes are expressed in the PVN [80,81,82], they are subjected to subtype-selective down- and up-regulation by agonists—for example, α_1D_ up-regulates while the α_1A_ and α_1B_ subtypes down-regulate in a concentration-dependent manner in face to agonist challenge (i.e., endogenous noradrenaline), and down-regulation of the latter was accompanied by reductions of mRNA [83], illustrating the complexity of in vivo regulation of α_1_-adrenoceptors, even in healthy animals. Since in the binding assay membranes from the entire hypothalamus were utilized, it is likely that a dilution effect of the α_1_-adrenoceptor subtypes expressed mainly in the PVN would occur, such as the alpha 1B and alpha 1D subtypes. Indeed, it has been shown that the mRNA for the α_1A_-adrenoceptor subtype is expressed at high levels almost in all hypothalamic nuclei [80], while distribution of the mRNA for the α_1B_-adrenoceptor subtype is mainly restricted to the PVN and the lateral hypothalamic nucleus [84]. In the PVN, α_1B_ adrenoceptor mRNA is highly expressed in CRH containing cells [81], a region also expressing high levels of adrenoceptor α_1D_ mRNA [82]. Given that [3H]-prazosin binding does not resolve among the three adrenoreceptor subtypes and, on the other hand, that both PVNs represent a minimal part of the total mass of the hypothalamus, the dilution effect should not significantly compromise the binding data of the total population of α_1_-adrenergic receptors in the whole hypothalamus. Is therefore difficult to reconcile the meaning of the [^3^H]-prazosin binding data from the whole hypothalamus with the expression data for the mRNA of the α_1A_-adrenoreceptor subtypes obtained with in situ hybridization, even more so if it is taken into consideration that the first technique measures α_1_-adrenergic receptor density located at the terminal neuronal sites where they migrate, i.e., in perikarya, dendrites and axon terminals, while the later one concerns the sites where α_1_-adrenergic receptors are produced, i.e., the neuronal somata.

Of note, higher α_1A_-adrenoceptor mRNA [85] and unchanged α_1A_, α_1B_ or α_1D_-adrenoceptor mRNA [86,87] levels in the PVN of animals submitted to different experimental models of chronic hypertension have been reported. Chronic stress sensitizes the HPA axis to further acute stress (as measured by transient plasma ACTH increase) in rats, altogether with enhancing the response to α_1_-adrenergic receptor activation in the PVN [64]. The second statement, the requirement of some kind of long-lasting sensitization at the PVN level to maintain hypertension and tachycardia at the long-term, was not approached in the present study. In this regard, it has been reported that noradrenaline induces α_1_-adrenoceptor-mediated increase and α_2_-adrenoceptor-mediated decrease on GABA-dependent spontaneous inhibitory postsynaptic current in a subset of parvocellular neurons of the PVN [88], which possibly represent a metaplastic regulation of GABAergic transmission in these neurons. Hippocampal long-term potentiation [89] and cerebral cortex long-term depression [90,91] have been reported to be promoted by α_1_-adrenoceptors, but studies on the neuroplasticity processes involved in long-lasting sensitization of the neurons of the PVN that promote sympathetic activation during chronic hypertension are still lacking. Of note, both early-life stress and early-life undernutrition similarly led to life-long alterations to the neuroendocrine stress system, partially by modifying epigenetic regulation of gene expression [92]. Increased CRF production via epigenetic mechanisms cannot be discarded but no epigenetic modifications underlying altered CRF expression in prenatally undernourished animals have been reported so far.

## 4. Materials and Methods

### 4.1. Animals

The experimental protocol and animal management followed the NIH Guide for the Care and Use of Laboratory Animals [93]. They were approved by the Committee for the Ethical Use of Experimental Animals (Protocol CBA_INTA #FR-2012-HP-01) at the Institute of Nutrition and Food Technology of the University of Chile. The experiments were carried out on male Wistar rats of 40 to 44 days of age from the inbred colony of the Institute of Nutrition and Food Technology, which were maintained in temperature-controlled conditions (22 ± 1 °C) under a 12:12 h inverted light–dark cycle (lights on from 20:00 to 08:00 h). All these animals were born from mothers submitted during pregnancy to one of the following nutritional conditions: (i) normal pregnant rats with free access to a 21% protein standard laboratory diet (Champion, Santiago, Chile) and (ii) undernourished pregnant rats with restricted access (10 g daily) to the standard laboratory diet throughout pregnancy. This amount of food is about 40% of the amount consumed by normal pregnant rats [25] and was given twice daily (5 g at 08:00 h and 5 g at 20:00 h) to minimize anxiety for feeding in the food-restricted pregnant dams. The dietary paradigm started one day before mating and continued throughout pregnancy. At birth, all pups were weighed, and litters were culled to eight male pups. The offspring prenatally undernourished were fostered at birth to well-nourished dams giving birth on that day to prevent postnatal undernutrition. The offspring born from eutrophic mothers were also fostered to well-nourished dams to equalize other factors that may depend on the rearing conditions in both groups (e.g., stress due to cross-fostering). During the lactation period, all dams received the standard laboratory diet ad libitum. After weaning at 22 days of age, the pups were housed eight per cage and fed on the standard laboratory diet. The body weights of pregnant mothers and pups were measured daily.

### 4.2. Drugs and Microinjection Procedures

The drugs administered were the α_1_-adrenoceptor agonist phenylephrine hydrochloride, the α_1_-adrenoceptor antagonist prazosin hydrochloride, the corticotropin-releasing factor (CRF), and the corticotropin-releasing factor antagonist α-helical CRF (9–41 fragment), all from Sigma-Aldrich (St. Louis, MO, USA). They were dissolved in artificial cerebrospinal fluid (aCSF), made of the following constituents: 7.46 g NaCl, 0.19 g CaCl_2_ (anhydrous), 0.20 g KCl and 0.20 g MgCl_2_, in 1.0 L of distilled water.

At 40 to 44 days of age, between 09:00 and 11:00 h, both normal eutrophic and undernourished rats were anesthetized with 50 mg/kg i.p. sodium pentobarbital and placed in a stereotaxic apparatus (Narishige ST-7, Narishige Scientific Instrument Lab., Tokyo, Japan) to microinject either intra-PVN or intra-LC, the agonists, antagonists, or aCSF. Then, the skull was exposed, and both the right and left PVNs were approached through 1.5-mm burr holes that were drilled bilaterally into the parietal bone, at coordinates A: −1.8 from bregma, L: 0.5 from the midline, and V: −7.4 from the cortical surface, in mm, while both LCs were bilaterally approached through 1.5-mm burr holes drilled bilaterally into the occipital bone, at coordinates A: −9.8 from bregma, L: 1.5 from the midline, and V: −5.5 from the cerebellar surface, in mm [49]. Two glass micropipettes with 20- to 30-μm tips, made by pulling borosilicate glass tubing of 1.0-mm outer diameter on a micropipette puller, were bilaterally advanced to the right and left nuclei to injecting 0.05 μL of either vehicle (aCSF) or the agonists/antagonists. The amount of drug microinjected in each nucleus was taken from previous studies [63,94]. The PVN received 6 nmol/side phenylephrine or 130 pmol/side prazosin, while the LC received 20 pmol/side CRF or 26 pmol /side α-helical CRF, both in normal (*n* = six in each group) and undernourished (*n* = six in each group) rats. Bilateral microinjections either in PVN or LC were simultaneously performed over 2 min with two electric-driven micro-infusion pumps (Stoelting Co., Wood Dale, IL, USA) using two 5-μL 7000 series Hamilton syringes connected to the micropipettes via PE10 catheters. In additional experiments (*n* = six in each group), the bilateral microinjection of an agonist into a nuclei pair (i.e., CRF intra-LC or phenylephrine intra-PVN) was preceded by a prior (10 min before) bilateral microinjection of the corresponding antagonist into the other, complementary nuclei pair (prazosin intra-PVN or α-helical CRF intra-LC), in order to study if the cardiovascular effects associated to neuronal activation in the LC can be counteracted by inactivating PVN neurons, or vice versa. Once the drugs were microinjected and the experiment was finished, the micropipettes were lifted and refilled with 0.5% Evans blue dye in saline, then relocated to the original stereotactic coordinates for the nuclei, and 0.05 µL of the dye was bilaterally microinjected for 2 min, according to a procedure already described [95]. Afterward, the animals were euthanized with an overdose of 100 mg/kg i.p. of sodium pentobarbital and perfused transcardially with 50 mL of 10% buffered formaldehyde. The brain was removed and processed histologically to verify the sites of microinjection with cresyl-violet staining (Figure 8).

### 4.3. Systolic Pressure and Heart Rate

Systolic pressure and heart rate were recorded from the tail of 40 to 44 day-old rats under pentobarbital anesthesia (50 mg/kg i.p.) employing a non-invasive blood pressure system (tail-cuff plethysmography, XBP 1000 Kent Scientific apparatus, Torrington, CT, USA) before and 10, 20, 30, 40 and 50 min after aCSF or drug microinjection into the PVN or the LC. Systolic pressure was expressed as mmHg, and heart rate in beats/min. As reported elsewhere [96], unlike isoflurane, or chloralose-urethane, or ketamine-xylazine, pentobarbital anesthesia had only a modest influence on blood pressure level and its maintenance by vasoactive systems (i.e., renin-angiotensin, nitric oxide, and sympathetic nervous system).

### 4.4. α_1_-Adrenoceptor Binding Assay

#### 4.4.1. Membrane Preparation

Experiments were carried out in six normal and six prenatally undernourished rats of 40 days of age. Animals were killed by decapitation, and their brain rapidly removed, weighed, and cooled on ice. The whole hypothalamus was excised, and the tissue was homogenized 1:8 (wt/vol) in ice-cold buffer (4 °C, 5 mM Tris–HCl, 5 mM EDTA, pH 7.4). Homogenates were then centrifuged at 1000× *g*, and the resulting supernatants were centrifuged at 30,000× *g* for 10 min at 4 °C. This step was repeated twice, and the resulting pellet was then resuspended in a buffer consisting of 50 mmol/L Tris–HCl, 0.5 mM EDTA, pH 7.4.

#### 4.4.2. Binding Assay for α_1_-Adrenoceptors

The density of the total α_1_-adrenoceptor population was assayed using [^3^H]-prazosin as the labeled ligand (Amersham, Chicago, IL, USA), which binds, almost equally, the α_1A_, α_1B_, and α_1D_ adrenoceptor subtypes [60]. Aliquots of membrane suspensions (0.1 mL) were diluted to a final volume of 0.5 mL in the assay buffer (50 mmol/L Tris–HCl, 0.5 mM EDTA, pH 7.4) in the presence of 0.1 to 45 nmol/L of [^3^H]-prazosin (83 Ci/mmol), and then incubated at 25 °C for 45 min. The reaction was terminated by adding 2 mL of ice-cold assay buffer, followed by rapid filtration under reduced pressure through Whatman GF/B glass fiber filters pre-washed with assay buffer. The tubes and filters were rapidly washed with assay buffer (four times with 2 mL), and the radioactivity was counted as automatically quench-corrected DPM in a Packard 1600TR liquid scintillation counter. Specific binding was defined as the amount of total bound radioactivity minus observed in the presence of a 1 μm excess of unlabeled prazosin. Dissociation constant (Kd) values and the maximum number of specific binding sites (Bmax) were calculated from the Scatchard’s plot-transformed binding data with iterative mass action law-based curve-fitting program LIGAND and expressed as pM and fmol/mg protein, respectively. Protein content was determined using the Bradford’s method [97] with bovine serum albumin as standard.

### 4.5. Non-Isotopic In Situ Hybridization for α_1_-Adrenoceptor mRNA Expression

Non-isotopic in situ hybridization was performed as described previously [98]. At 40 days of age, six normal and six prenatally undernourished rats were perfused with physiological saline followed by 4% paraformaldehyde in 0.1 M phosphate buffer, pH 7.4, and the brains were removed and maintained in 20% sucrose for 72 h. After that, the brains were cut into blocks and kept in a paraformaldehyde fixation solution for two hours. The blocks were then transferred and maintained in 20% sucrose in a 0.1 M phosphate buffer for 12 h. Finally, the brain blocks were sectioned into 30 µm coronal slices in a cryostat.

For in situ hybridization, the following deoxynucleotide probe for the α_1A_-adrenoceptor mRNA (NAdra1A, from Integrated DNA Technologies Inc., Coralville, IA, USA) was used: 5′-GGA GCT GGT GGG TGG GTG CAG TTG GAG CCT TCC GAA GCA TTT TCA-3′, which was 3′ end-labeled with digoxigenin. According to in situ hybridization experiments by Day et al. [99], high expression levels of α_1A_-adrenoceptor mRNA are found in the rat PVN, while mRNA for the α_1B_ and the α_1D_-adrenoceptor subtypes are expressed at low and very low levels in the same region, respectively. Thirty-micrometer coronal slices were hybridized overnight, at 40 °C for 16 h, in the presence of 4 pmol/mL of the α_1_-adrenoceptor digoxigenin-labeled probe. After hybridization, tissue slices were rinsed twice with saline sodium citrate and once in saline sodium citrate for 10 min each at 42 °C. For control experiments, the tissue slices were incubated with the DIG-labeled probe in the presence of an excess (100×) of an unlabeled probe. The digoxigenin label was detected with anti-digoxigenin antibody conjugated to alkaline phosphatase (Boehringer Mannheim GmbH Biochemica, Mannheim, Germany), using nitroblue tetrazolium and 5-Bromo-4-chloro-3-indolyl phosphate (Invitrogen) as enzyme substrates. Finally, slices were mounted on glass slides in 0.1% gelatin and observed under light microscopy. Microphotographs at 10× and 20× were transformed into 8-bit format and pixels quantified employing the Image J software.

### 4.6. Statistical Analysis

The results obtained were expressed as mean values ± SEM. All statistical analyses were performed with the GraphPad Prism software (GraphPad Software, Inc., San Diego, CA, USA). For the effects of dietary treatments on body and brain weights, for binding assays, and for in situ hybridization, intergroup comparisons were made using a two-tailed unpaired Student’s *t*-test. Analysis of intra- and intergroup comparisons of time–course changes in systolic blood pressure and heart rate was carried out by two-way repeated measures ANOVA followed by the Dunnett’s multiple comparisons (intragroup comparisons) or the Bonferroni’s multiple comparisons post hoc tests. Significance was accepted at an alpha level of 0.05.

## 
5. Conclusions


In conclusion, data presented herein support the following: (1) simultaneous concurrent tonic neuronal activity in the PVN and the LC is required to maintain elevated scores of systolic pressure and heart rate in prenatally malnourished animals, as revealed by the fact that either antagonist independently alleviated both hypertension and tachycardia in these animals; (2) functional reciprocal excitatory noradrenergic and CRFergic connections between the PVN and the LC are involved in the generation of hypertension and increased heart rate in previously malnourished adult rats, as revealed by the fact that an agonist that was ineffective to produce cardiovascular effects in these animals when injected to the appropriate nucleus recovered full efficacy after inactivation of the complementary nucleus with the suitable antagonist; (3) desensitization of PVN α_1_-adrenoceptors does not occur in undernourished rats, as revealed by the fact that early undernutrition did not modify the number of α_1_-adrenoceptor binding sites in the hypothalamus as related to eutrophic controls, thus allowing the PVN to operate as the output locus in the paraventricular–coerulear loop; (4) both increased blood pressure and heart rate induced in healthy normotensive rats by either α_1_-adrenoceptor-mediated excitation of PVN neurons or CRF receptor-mediated excitation of LC neurons did not imply serial or reciprocal excitatory interactions between the two nuclei, as revealed by the fact that the cardiovascular effects observed were not prevented by disruption of the communication between the nuclei with an appropriate antagonist.

Therefore, our results support the central hypothesis that tonic reciprocal excitatory connections between the PVN and the LC help to explain the hypertension and increased heart rate observed in prenatally undernourished adult rats. Further work is required to explain the underlying mechanisms that may account for the differences in tonic activity of the LC-PVN loop between prenatally undernourished and eutrophic subjects, probably related to a loss of inhibition in one or both nuclei.

## Figures and Tables

**Figure 1 molecules-26-03568-f001:**
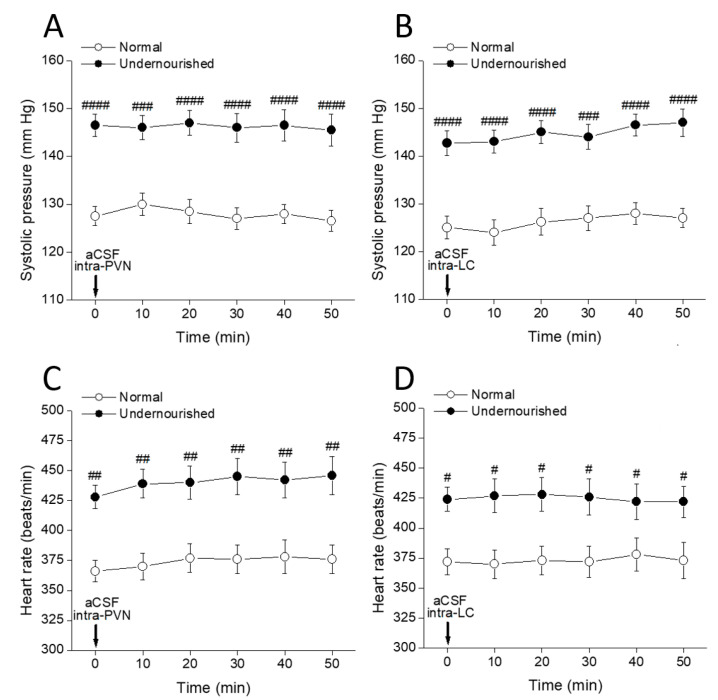
Time-course of changes in systolic pressure (**A**,**B**) and heart rate (**C**,**D**) of normal and prenatally undernourished rats after intra-PVN or intra-LC microinjection (arrow at time 0 min) of artificial cerebrospinal fluid (aCSF). Values are means ± SEM, *n* = six rats in each group. Two-way repeated measures ANOVA was used to identify the nutritional treatment and/or the time elapsed after aCSF microinjection as significant factors in the effect examined ((**A**): for the effect of intra-PVN aCSF on systolic pressure, F_nutrition(1, 10)_ = 161.8 and F_time(5, 50)_ = 0.1753; (**B**): for the effect of intra-LC aCSF on systolic pressure, F_nutrition(1, 10)_ = 61.07 and F_time(5, 50)_ = 0.4028; (**C**): for the effect of intra-PVN aCSF on heart rate, F_nutrition(1, 10)_ = 224.6 and F_time(5, 50)_ = 0.5895; (**D**): for the effect of intra-LC aCSF on heart rate, F_nutrition(1, 10)_ = 33.68 and F_time(5, 50)_ = 0.0164). Dunnett’s multiple comparisons test detected no significant intragroup differences at any time after aCSF (referred to the pre-aCSF control at time = 0 min), while Bonferroni’s multiple comparisons test detected significant intergroup differences at all times between normal and undernourished animals (**^#^**
*p* < 0.05, **^##^**
*p* < 0.01, **^###^**
*p* < 0.001, **^####^**
*p* < 0.0001).

**Figure 2 molecules-26-03568-f002:**
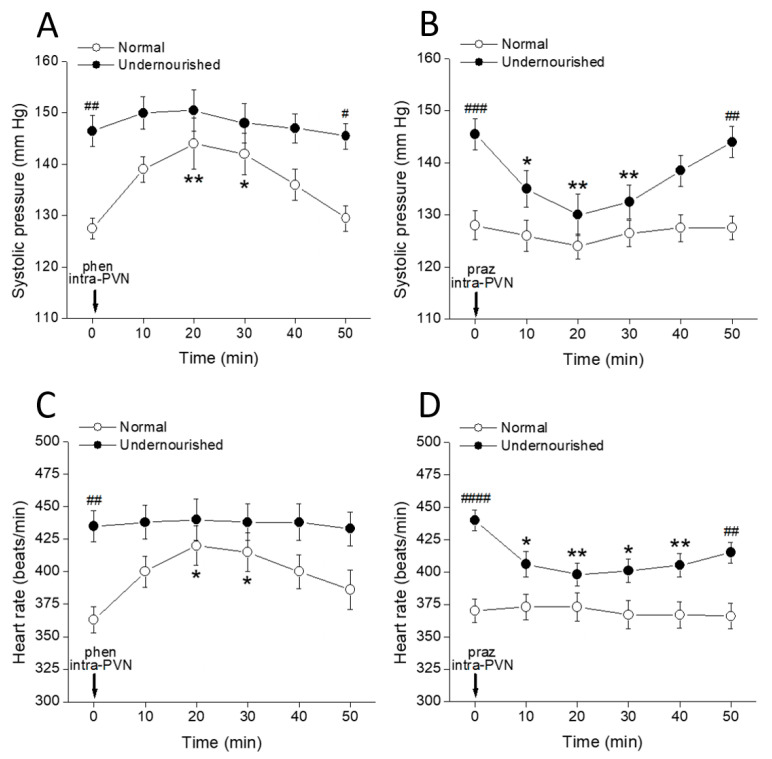
Time-course of changes in systolic pressure (**A**,**B**) and heart rate (**C**,**D**) of normal and prenatally undernourished rats after intra-PVN microinjection (arrow at time 0 min) of the α_1_-adrenoceptor agonist phenylephrine (phen, 6 nmol/side; (**A**,**C**)) or the α_1_ adrenoceptor antagonist prazosin (praz, 130 pmol/side; (**B**,**D**)). Data are mean values ± SEM, *n* = six rats in each group. Two-way repeated measures ANOVA was used to identify the nutritional treatment and/or the time elapsed after drug microinjection as significant factors in the effect examined ((**A**): for the effect of phenylephrine on systolic pressure, F_nutrition(1, 10)_ = 53.40 and F_time(5, 50)_ = 3.066; (**B**): for the effect of prazosin on systolic pressure, F_nutrition(1, 10)_ = 25.43 and F_time(5, 50)_ = 3.595; (**C**): for the effect of phenylephrine on heart rate, F_nutrition(1, 10)_ = 21.69 and F_time(5, 50)_ = 1.457; (**D**): for the effect of prazosin on heart rate, F_nutrition(1, 10)_ = 30.13 and F_time(5, 50)_ = 1.879). Significance symbols arising from the post hoc tests indicate either intragroup differences after phenylephrine or prazosin referred to the pre-drug control at time = 0 min (* *p* < 0.05, ** *p* < 0.01, Dunnett’s multiple comparisons test) or intergroup differences between normal and undernourished subjects (**^#^**
*p* < 0.05, **^##^**
*p* < 0.01, **^###^**
*p* < 0.001, **^####^**
*p* < 0.0001, Bonferroni’s multiple comparisons test).

**Figure 3 molecules-26-03568-f003:**
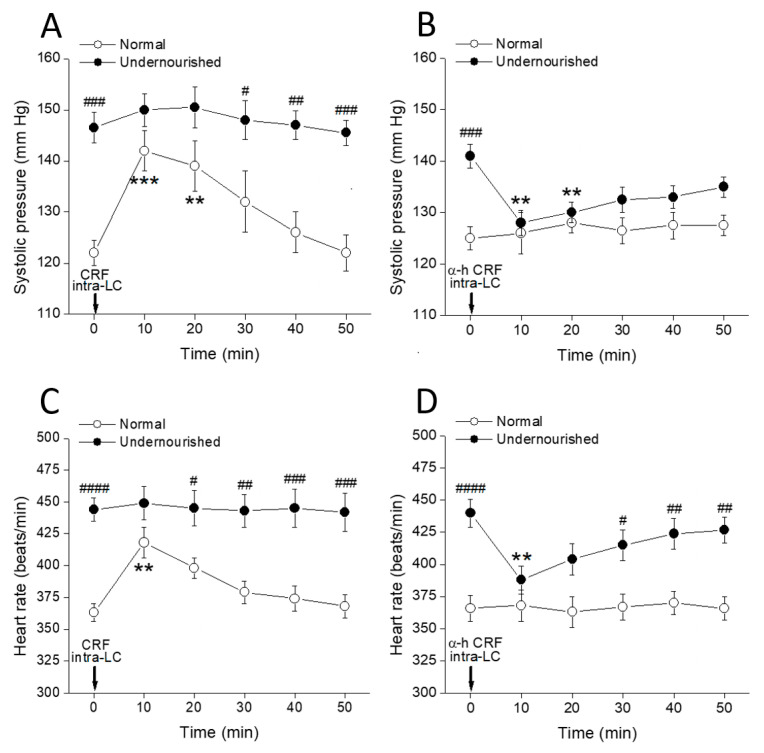
Time-course of changes in systolic pressure (**A**,**B**) and heart rate (**C**,**D**) of normal and prenatally undernourished rats after intra-LC microinjection (arrow at time 0 min) of CRF (20 pmol/side; (**A**,**C**)) or the CRF receptor antagonist α-helical CRF (α-h CRF, 26 pmol/side; (**B**,**D**)). Values are means ± SEM, *n* = six rats in each group. Two-way repeated measures ANOVA was used to identify the nutritional treatment and/or the time elapsed after drug microinjection as significant factors in the effect examined ((**A**): for the effect of phenylephrine on systolic pressure, F_nutrition(1, 10)_ = 30.58 and F_time(5, 50)_ = 4.772; (**B**): for the effect of prazosin on systolic pressure, F_nutrition(1, 10)_ = 17.40 and F_time(5, 50)_ = 1.441; (**C**): for the effect of phenylephrine on heart rate, F_nutrition(1, 10)_ = 61.82 and F_time(5, 50)_ = 2.161; (**D**): for the effect of prazosin on heart rate, F_nutrition(1, 10)_ = 46.11 and F_time(5, 50)_ = 1.684). Significance symbols arising from the post hoc tests indicate either intragroup differences after CRF or α-helical CRF referred to the pre-drug control at time = 0 min (** *p* < 0.01, *** *p* < 0.001, Dunnett’s multiple comparisons test) or intergroup differences between normal and undernourished subjects (**^#^**
*p* < 0.05, **^##^**
*p* < 0.01, **^###^**
*p* < 0.001, **^####^**
*p* < 0.0001, Bonferroni’s multiple comparisons test).

**Figure 4 molecules-26-03568-f004:**
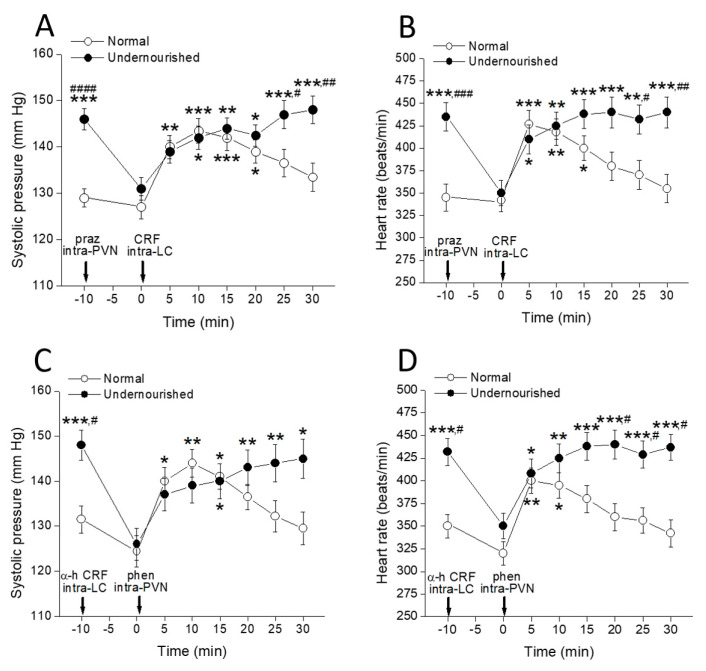
Time-course of changes in systolic pressure (**A**,**C**) and heart rate (**B**,**D**) of normal and prenatally undernourished rats after intra-PVN or intra-LC microinjection of phenylephrine (phen) or CRF (arrow at time 0 min), respectively, with or without pre-injection of an antagonist (praz or α-h CRF respectively, arrow at time −10 min) into the complementary nucleus. Values are means ± SEM, *n* = six rats in each group. Two-way repeated measures ANOVA was used to identify the nutritional treatment and/or the time elapsed after drug microinjection as significant factors in the effect examined ((**A**): effect on systolic pressure of CRF intra-LC preceded by prazosin intra-PVN, F_nutrition(1, 10)_ = 28.02 and F_time(7, 70)_ = 6.058; (**B**): effect on systolic pressure of phenylephrine intra-PVN preceded by α-helical CRF intra-LC, F_nutrition(1, 10)_ = 15.90 and F_time(7, 70)_ = 5.806; (**C**): effect on heart rate of CRF intra-LC preceded by prazosin intra-PVN, F_nutrition(1, 10)_ = 11.33 and F_time(7, 70)_ = 4.233; (**D**): effect on heart rate of phenylephrine intra-PVN preceded by α-helical CRF intra-LC, F_nutrition(1, 50)_ = 31.43 and F_time(7, 70)_ = 5.171). Significance symbols arising from the post hoc tests indicate either intragroup differences after drug treatment referred to the control at time = 0 min (* *p* < 0.05, ** *p* < 0.01, *** *p* < 0.001, Dunnett’s multiple comparisons test) or intergroup differences between normal and undernourished subjects (**^#^**
*p* < 0.05, **^##^**
*p* < 0.01, **^###^**
*p* < 0.001, **^####^**
*p* < 0.0001, Bonferroni’s multiple comparisons test).

**Figure 5 molecules-26-03568-f005:**
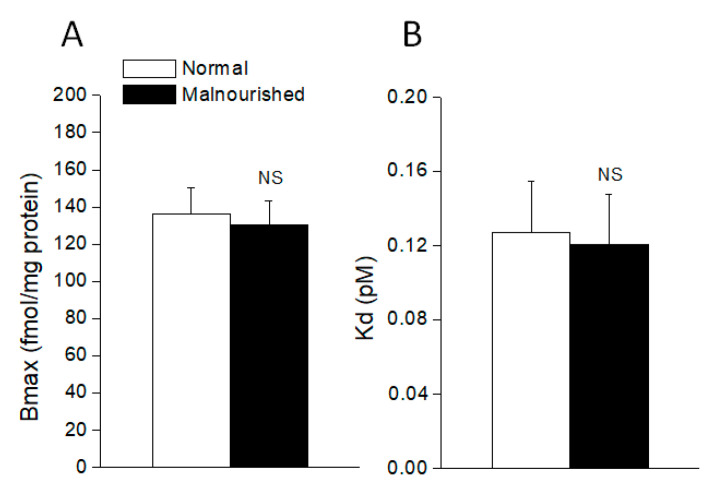
The number of total α_1_-adrenoceptor binding sites (Bmax) (**A**) and dissociation constant (Kd) (**B**) in the hypothalamus of normal and malnourished rats, as revealed by [^3^H]-prazosin binding assay. Values are the mean ± SEM of six eutrophics and six prenatally undernourished rats of 40 days of age, the difference between the two groups being not statistically significant (NS: not significant; for Bmax comparison *p* = 0.7730, *t* = 0.2964; for Kd comparison *p* = 0.8805, *t* = 0.1543; two-tailed unpaired Student’s *t*-test).

**Figure 6 molecules-26-03568-f006:**
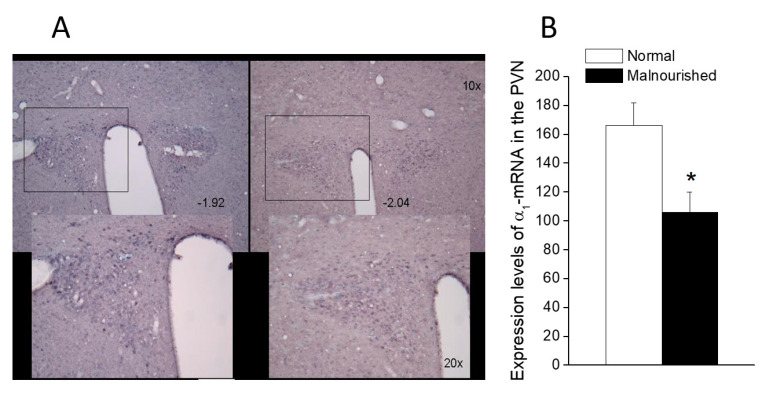
α_1A_-Adrenoceptor mRNA expression in the paraventricular (PVN) nucleus of normal and malnourished rats, as revealed by in situ hybridization. (**A**) Photomicrographs of 30-µm thick brain coronal sections passing at the level of the PVN of a normal rat (left) and an undernourished rat (right), showing in situ hybridization staining for α_1A_-adrenoceptor mRNA in the presence of the α_1A_-adrenoceptor digoxigenin-labeled probe. Photomicrographs correspond to sections located about 2.0 mm behind bregma, confirmed by comparison with Figures 49 (−1.92 from bregma) and 50 (−2.04 from bregma) of the rat brain atlas of Paxinos and Watson [49]. Pixel counting was performed within the areas enclosed by the 500 × 400 µm rectangles, which in the lower microphotographs are magnified ×2 concerning the upper ones (see scale bars). (**B**) In situ expression of α_1A_-adrenoceptor mRNA in the PVN of six eutrophics and six prenatally undernourished rats of 40 days of age, quantified by pixel counting. Values are the mean ± SEM. A comparison of data from the eutrophic and malnourished groups was made using a two-tailed unpaired Student’s *t*-test (* *p* = 0.0181, *t* = 2.822).

**Figure 7 molecules-26-03568-f007:**
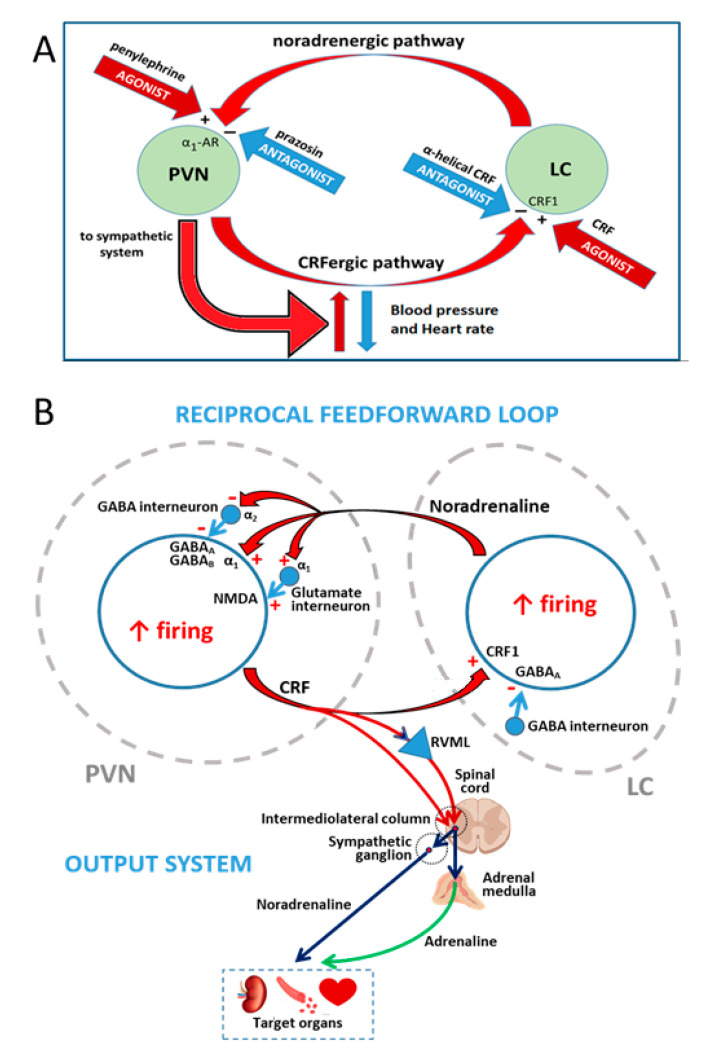
Summary of the working hypothesis, results and discussion. (**A**) Reciprocal excitatory interactions between the paraventricular nucleus of the hypothalamus (PVN) and the locus coeruleus (LC) led to hypertension and tachycardia in prenatally undernourished adult rats, increasing the outflow from the PVN to the sympathetic system. The closed loop works at a high activity level in prenatally undernourished adult rats and could be inhibited by the antagonists (−) prazosin (PVN) or α-helical CRF (LC), rescuing the animals from hypertension and tachycardia but it could not be overactivated by the agonists (+) phenylephrine (PVN) or CRF (LC). In eutrophic animals the loop works at a lower activity level that could not be inhibited by prazosin (PVN) or α-helical CRF (LC) but activated by phenylephrine (PVN) or CRF (LC), inducing hypertension and tachycardia. Activation of the loop (red arrows) led to hypertension and tachycardia in eutrophic animals, while interruption of the loop (blue arrows) led to normal blood pressure and heart rate in prenatally undernourished adult rats. (**B**) A more detailed view of the PVN-LC loop, taking into account the role of interneurons in the neural network of the PVN and the LC as supported by the literature: α_1_ and α_2_ receptors for noradrenaline, GABA_A_ and GABA_B_ receptors for γ-aminobutiric acid, NMDA receptor for glutamate, and CRF1 receptor for CRF are indicated in a CRF-expressing neuron within the PVN and a noradrenaline-synthesizing neuron within the LC; plus sign (+) and minus (−) sign represent receptor-mediated excitation or inhibition, respectively. The PVN output to target organs mediating hypertension and tachycardia via the sympathetic system is depicted; RVLM, rostral ventrolateral medulla.

**Figure 8 molecules-26-03568-f008:**
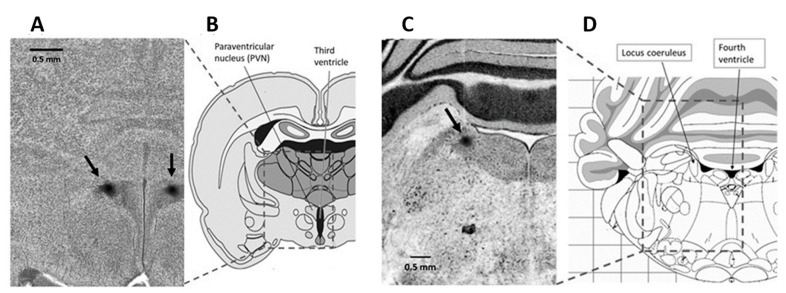
Photomicrographs of rat brain coronal, transverse sections (scale bar = 0.5 mm) showing stereotaxic dye labeling (see arrows) in the PVN (**A**) and the LC (**C**), alongside a diagrammatic representation of rat brain sections allowing to situate the PVN and the LC in specific regions of the hypothalamus (**B**) and the brainstem (**D**), respectively.

**Table 1 molecules-26-03568-t001:** Body weight, brain weight, systolic pressure, diastolic pressure, and heart rate of normal and malnourished rats at 40 days of age.

	Normal Group (*n*)	Undernourished Group (*n*)
Body weight (g)	153.6 ± 2.8 (36)	139.8 ± 3.3 ** (36)
Brain weight (mg)	1374.8 ± 9.3 (36)	1292.1 ± 16.5 *** (36)
Systolic pressure (mm Hg)	129.3 ± 2.9 (18)	144.9 ± 4.1 ** (18)
Diastolic pressure (mm Hg)	84.9 ± 5.9 (18)	88.1 ± 7.7 ^NS^ (18)
Heart rate (beats/min)	353.5 ± 19.0 (18)	430.1 ± 30.4 * (18)

Values are means ± SEM. Parentheses enclose number of rats. Significance of difference (two-tailed unpaired Student’s *t*-test) between normal and malnourished rats: * *p <* 0.05, ** *p <* 0.01, *** *p <* 0.001, NS: not significant.

## Data Availability

Data available upon request.

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
