# Peer review of "Hypertension in Prenatally Undernourished Young-Adult Rats Is Maintained by Tonic Reciprocal Paraventricular–Coerulear Excitatory Interactions"

_molecules, 2021, doi:10.3390/molecules26123568_

Round 1

Reviewer 1 Report

 The main aim of this article is to study if the previously demonstrated hypertension developed after undernutrition of the mother during gestation, is maintained by neuronal interactions between locus coeruleus and paraventricular nucleus. This focus is important to apply fundamental cellular mechanisms of neuronal function to unveil the changes that could be responsible of diseases in the adulthood derived from the environmental conditions to which the gestating mother was exposed.  There was an interesting set of experiments and physiological methodologies to obtain many data of relevance.

The most difficult part of the work was to understand the limits between the data already published by the group or by others, with the new data presented in the paper. How the new data are adding new understanding of the central circuits controlling blood pressure in under nutrition during gestation. The title of the work postulate that: “Hypertension in prenatally-undernourished young-adult rats is maintained by tonic reciprocal paraventricular-coerulear excitatory interactions”, thus the tonic reciprocal interaction must be presented in a clear form so not to be confused with many other data of the same group.

Main modifications:

a.- It is important to clearly present the previous knowledge that support the reciprocal connection between locus coeruleus and paraventricular nucleus, by the noradrenergic and CRHergic neurons, which are the receptors present  and their location. With these data it could be more clear to understand the rationale of the administration of alfa agonist and antagonist, the same as the agonist and antagonist for CRH.

b.- Are there studies showing the neurons in which are located the adrenergic and peptidergic receptors?. Are these receptors close to the respective neurons? If so, it could be more easy to discuss the data.

c.- It is not discussed if the receptors are located in the reciprocal neurons studied or there are other interneurons in the LC or PVN (most probably there are!) thus to focus the discussion.

d.- Is it considered to have only positive interactions or there are some negative effect on the neuronal circuit?

e.- It could be very helpful to include a diagram in which to draw the neurons and their connection between the LC and PVN.

f.- With a diagram, it could be clearer to understand how the same effect either in the LC or in the PVN produced the same effect on systolic pressure. How the interaction in the LC and PVN are related with the final output signal that is conducted by sympathetic nerves to the auricular pacemaker to change the frequency and systolic pressure.

Author Response

Cover Letter:

We thank the reviewers for the constructive comments about our manuscript submitted to Molecules, which significantly improved substance and form of the manuscript. We take into consideration all the comments and suggestions raised by the reviewers, and all the modifications we made in the new version of the manuscript are highlighted using the “track changes” function in Microsoft Word. All new references appearing as consequence of the review process were added in the text of the new manuscript, and the reference list was rearranged accordingly.

Next, we explain point-by-point all the revisions made in the manuscript:

Reviewer 1

Main modifications:

a.- It is important to clearly present the previous knowledge that support the reciprocal connection between locus coeruleus and paraventricular nucleus, by the noradrenergic and CRHergic neurons, which are the receptors present  and their location. With these data it could be clearer to understand the rationale of the administration of alfa agonist and antagonist, the same as the agonist and antagonist for CRH.

Response: Two short paragraphs addressing the points stated by the reviewer are now provided in the new version of manuscript. The first paragraph, inserted in the Introduction section (page 3, first paragraph, lines 3 to 14), concerns to the receptors involved, where they are present and their location. The second paragraph, also inserted in the Introduction section (pag3, second paragraph, lines 2 to 8), is devoted to the reviewer concern regarding how the new data are adding new understanding of the central circuits controlling blood pressure in undernutrition during gestation.

b.- Are there studies showing the neurons in which are located the adrenergic and peptidergic receptors? Are these receptors close to the respective neurons? If so, it could be easier to discuss the data.

Response: Yes, there are some studies that could clarify these aspects. This was answered as part of question “a.” (see above), as a short paragraph added to the Introduction section (page 3, first paragraph, lines 3 to 14) devoted to clarify in which neurons are located the adrenergic and peptidergic receptors involved in the reciprocal feedforward circuit. This is complemented by a new paragraph inserted in Discussion section (page 14, second paragraph, then throughout page 15 until line 16).

c.- It is not discussed if the receptors are located in the reciprocal neurons studied or there are other interneurons in the LC or PVN (most probably there are!) thus to focus the discussion.

Response: The reviewer is right. We did not addressed this topic in the original manuscript. Consequently, a fair paragraph addressing these issues is now provided in the Discussion section of the new manuscript (page 14, second paragraph, then throughout page 15 until line 16).

d.- Is it considered to have only positive interactions or there are some negative effects on the neuronal circuit?

Response: The paragraph inserted in the Discussion section regarding the organization and function of local neural circuits in both the PVN and the LC (page 14, second paragraph, then throughout page 15 until line 16, results from question “c.” of the reviewer) may shed light on these aspects. On the other hand, the data provided in the manuscript indicate that in previously undernourished animals, a positive feedforward interaction between the PVN and the LC is observed as net effect, while no such a positive interaction can be seen in eutrophic controls. Each of the two states (positive loop ‘on’ or ‘off’) likely is the result of a complex interplay between excitatory an inhibitory actions occurring locally both at the PVN and LC, in the context of the so-called heterosynaptic modulation, where tonic reciprocal feedforward excitation between these nuclei emerges in undernourished animals likely as a consequence of fetal programming, while this not happens in eutrophic subject. Today, there is no sufficient information to dissect such the neural bases of such a behavior, so we declare at the end of the Conclusion section “Further work is required to explain the underlying mechanisms that may account for the differences in tonic activity of the LC-PVN loop between prenatally undernourished and eutrophic subjects, probably related to a loss of inhibition in one or both nuclei”.

e.- It could be very helpful to include a diagram in which to draw the neurons and their connection between the LC and PVN.

Response: Done. A new figure (Figure 7, page 16 of the new version of the manuscript) was included in the Discussion section drawing the neurons and connections of the circuit.

f.- With a diagram, it could be clearer to understand how the same effect either in the LC or in the PVN produced the same effect on systolic pressure. How the interaction in the LC and PVN are related with the final output signal that is conducted by sympathetic nerves to the auricular pacemaker to change the frequency and systolic pressure.

Response: Done. The new Figure 7 (page 16 of the new version of the manuscript) includes the output connection from the PVN to target organs -including heart- through the sympathetic system.

Reviewer 2 Report

This is a very interesting and important paper that addresses the bidirectional regulation of the paraventricular nucleus with the locus cereuleus. The data suggests that there is an internal loop regulation with an outcome in systolic regulation. Furthermore, early malnutrition, a clinically important problem, is addressed using an elegant rat model.

There are some minor concerns:

  1. Statistics. Please consult a statistician. It seems that an ANOVA with repeated measure would be more appropriate for a number of the datasets (Figure 1-4).
  2. The study is conducted under pentobarbitol anesthesia. Since pentobarbitol works through the GABA-A pathway, a more extensive possible interaction of anesthetic with the outcome effect.
  3. Please better justify why the doses administered were chosen. Was it based on pharmacologic characteristics and/or previous studies? It should also be expressed as moles rather than weight.
  4. In the binding assay, membranes from the entire hypothalamus was utilized. Provide more discussion of what this means since there may be a dilution effect - PVN vs entire hypothalamus. 

Author Response

Cover Letter:

We thank the reviewers for the constructive comments about our manuscript submitted to Molecules, which significantly improved substance and form of the manuscript. We take into consideration all the comments and suggestions raised by the reviewers, and all the modifications we made in the new version of the manuscript are highlighted using the “track changes” function in Microsoft Word. All new references appearing as consequence of the review process were added in the text of the new manuscript, and the reference list was rearranged accordingly.

Next, we explain point-by-point all the revisions made in the manuscript:

Reviewer 2

There are some minor concerns:

  1. Statistics. Please consult a statistician. It seems that an ANOVA with repeated measure would be more appropriate for a number of the datasets (Figure 1-4).

Response: The reviewer is right. Therefore, we did the statistical analysis in Figures 1 to 4 of the new version of the manuscript, using two-way repeated measures ANOVA. The analysis did not modify neither results nor conclusions.

  1. The study is conducted under pentobarbitol anesthesia. Since pentobarbitol works through the GABA-A pathway, a more extensive possible interaction of anesthetic with the outcome effect.

Response: We used the recommendation of Bencze et al. 2013, as stated (and referenced) in Materials and Methods section of the manuscript. In the study by Bencze et al. 2013 the authors compared the effect of four widely used anesthetics (pentobarbital, isoflurane, ketamine-xylazine and chloralose-urethane) on the participation of the sympathetic nervous system, the renin-angiotensin system, and nitric oxide synthesis in the maintenance of blood pressure, in anesthetized either normal (Wistar) or spontaneously hypertensive rats, which is very consistent with our goals in the present manuscript (where we also used Wistar rats). They found that the interference of pentobarbital anesthesia with cardiovascular experiments is smaller, as compared to the other anesthetics used, in both rat strains (Wistar and spontaneous hypertensive) under all the conditions studied. Other studies have reported that ketamine/xylazine combination almost invariably results in significant hypotension (Middleton et al., 1982; Allen et al., 1986); in addition, in all cases it will not reliably reach surgical anesthesia, and it can also cause profound cardiac depression, as revealed by decreased heart rate, decreased cardiac output, and hypotension; for surgery longer than 20 minutes, animals will likely require additional anesthetic (LARC Veterinarians’ Anesthesia and Analgesia Recommendations for UCSF Laboratory Animals, 2018). Neuroleptanalgesic combinations (a potent opioid, usually fentanyl, plus a tranquillizer/sedative) produce severe respiratory depression, and the rapid and complete reversal of fentanyl anesthesia could be inconvenient for interventions lasting more than 15 min (Paul Flecknell, Laboratory Animal Anesthesia, Third Edition, 2009). Propofol produces a moderate fall in systolic blood pressure and a small fall in cardiac output (Sebel and Lowdon, 1989) but causes significant respiratory depression in most species (Glen, 1980), and it is therefore advisable to provide supplemental oxygen; in addition, propofol must be given intravenously and not intraperitoneally. Similarly, chloral hydrate has moderate effects on the cardiovascular system and on baroreceptor reflexes, but it can produce severe respiratory depression; together with this, there is considerable strain variation in the response to chloral hydrate in rodents (Paul Flecknell, Laboratory Animal, Third Edition, 2009). Alpha-chloralose produces stable, long-lasting but light anaesthesia, associated with minimal cardiovascular and respiratory system depression (Holzgrefe et al., 1987; Svendsen et al., 1990) but also with a very poor analgesic effect; so it may be useful for providing long-lasting light anaesthesia for procedures involving no painful surgical interference, which is not the case of studies reported in the present manuscript where surgery involved osteotomy and dissection of skin and muscle. Urethane resembles chloralose in producing long-lasting and a higher degree of analgesia, but it is also a carcinogen, so its use should be avoided whenever possible (Paul Flecknell, Laboratory Animal Anesthesia, Third Edition, 2009). Though superseded in many applications by newer anesthetics, barbiturates still have their place in the animal laboratory. Sodium pentobarbital is most frequently used in terminal or acute studies, like those described in the present manuscript; animals do not feel pain when the drug is at a surgical plane of anesthesia (50 mg/kg i.p. for the rat), as reported by the LARC Veterinarians’ Anesthesia and Analgesia Recommendations for Laboratory Animals, University of California at San Francisco, 2018, in their Web page. On these bases, we decided to utilize sodium pentobarbital as anesthetic agent, as recommended by Bencze et al., 2013, for cardiovascular physiopharmacological studies. In the new version of the manuscript, we shortly expand the explanation of why we used pentobarbital anesthesia (page 19, paragraph after Figure 7, lines 5 to 9).

  1. Please better justify why the doses administered were chosen. Was it based on pharmacologic characteristics and/or previous studies? It should also be expressed as moles rather than weight.

Response: We changed the amount of drug microinjected into each PVN or each LC, from a weight basis to moles (Material and Methods section, page 18, third paragraph, lines 14 to 16 within the paragraph, in the legends of Figures 2 and 3 in pages 6 and 7 respectively, and throughout the manuscript, where appropriate). Concerning the amount of drug microinjected in each nucleus, this was taken from previous studies (Hwang et al. 1998 and Konishi et al. 1993), which is declared in the Material and Methods section (page 18, third paragraph, lines 14 to 16 within the paragraph of the new version of the manuscript). Into the PVN, Hwang et al. microinjected bilaterally 10 nmol phenylephrine or 100 pmol prazosin in 50 nl volume, while we microinjected bilaterally 6 nmol phenylephrine or 130 pmol prazosin, in a similar volume of 50 nl. Into the LC, Konishi et al. microinjected bilaterally 20 pmol of CRF or 26 pmol of α-helical CRF(9-41), in a 0.5 µl volume, whereas we microinjected the same amounts of CRF and α-helical CRF(9-41), but in a smaller volume of 50 nl. These amounts are also in a similar range to those microinjected by Okada and Yamaguchi, 2017 (5 and 10 nmol phenylephrine intra PVN), Ma and Morilak, 2005 (2 to 50 nmol phenylephrine intra PVN), Shih et al. 1995 (50 pmol prazosin intra PVN), Ciccocioppo et al., 2003 (20 pmol CRF intra LC), Ku et al., 1998 (40 pmol CRF intra LC).

  1. In the binding assay, membranes from the entire hypothalamus was utilized. Provide more discussion of what this means since there may be a dilution effect - PVN vs entire hypothalamus.

Response: The reviewer is right. We declared in the Discussion section of the manuscript (page 16; third paragraph, lines 12 to 15) that the [3H]-prazosin binding assay performed did involve tissue from the entire hypothalamus, a technical limitation originated in the amount of tissue required for the determination, but we did not discuss the implications of such methodological problem. Consequently, a paragraph addressing this question is now provided in the Discussion section of the new manuscript (page 17, first paragraph, lines 4 to 22).

Round 2

Reviewer 1 Report

Thanks to the authors. All my previous comments were adequately responded